# The Importance of the Outdoor Environment for the Recovery of Psychiatric Patients: A Scoping Review

**DOI:** 10.3390/ijerph20032240

**Published:** 2023-01-27

**Authors:** Mikkel Hjort, Martin Mau, Michaela Høj, Kirsten K. Roessler

**Affiliations:** 1Department of Psychology, University of Southern Denmark, 5230 Odense, Denmark; 2Health, Social Work and Welfare Research, UCL University College, 5230 Odense, Denmark; 3Mental Health Center Ballerup, Competence Center for Rehabilitation and Recovery, 2750 Ballerup, Denmark

**Keywords:** recovery, rehabilitation, mental health, well-being, therapeutic landscapes, gardening, nature, psychiatric

## Abstract

(1) Background: Research has shown that patients with mental health diagnoses experience less anxiety and depressive symptoms and higher levels of ‘well-being’ when they spend time in natural environments as part of their treatment. It has been suggested that there is a relationship between the outdoor settings and the recovery of psychiatric patients. Recovery describes an individual process, which can vary from person to person. (2) Methods: This scoping review examined the relationship between the physical environment and the recovery of psychiatric patients. Systematic searches in three online databases, namely Medline, Embase, and PsycINFO, were performed using a selection of psychiatric, environmental, and recovery terms and included both quantitative and qualitative studies. In general, ‘well-being’ serves as an overarching indicator when it comes to research on how outdoor settings can affect mental health. Well-being was expressed in terms of mood, social relations, and autonomy. (3) Results: A total of 8138 records were screened, 85 studies were included for full-text reading, and five articles were included in the final analysis. The review showed in general that outdoor settings, more specifically gardening, contact with nature, and a safe environment can be related to the well-being of patients on psychiatric wards. (4) Conclusions: The five reviews allow us to conclude that outdoor settings can be seen as a comprehensive resource for mental health.

## 1. Introduction

The outdoor environment is providing a wide range of benefits for physical and mental health [1,2,3]. There has recently been increasing focus on the link between spending time in natural environment and well-being [4,5,6], but a review study from 2019 revealed a paucity of research into environment suited specifically to recovery among psychiatric patients [7]. In addition, research indicates that patients with stress-related problems, such as burn-out, depression, and anxiety, may experience shorter periods of illness, fewer symptoms of anxiety and depression, and a higher degree of well-being if they spend time or work in outdoor environments as part of the treatment [8,9,10,11,12]. This is in line with the stress reduction theory (SRT) proposed by Roger Ulrich in 1981. SRT focuses on how natural environments can reduce physiological stress and aversive emotion. The theory proposes that natural environments promote recovery from stress, while urban environments tend to hinder the same process [13].

A review focusing on Nordic experiences (Norway, Sweden, Finland, Denmark and Iceland) from 2015 further describes how the green care concept (the use of animals, plants and nature actively to promote health) can provide activities for people with mental health difficulties (among other groups) and strengthen mastery, coping, and meaningfulness [14]. All of these are important elements in the process of *personal* recovery, further elaborated below.

Personal recovery involves an individual process and can vary from person to person. In contrast to clinical recovery, personal recovery focuses on the individual’s subjective process of finding a path towards living a meaningful life, rather than focusing on the absence of disease symptoms [15]. The process is rarely linear or dependent on a programme of formal treatment. Instead, factors, such as positive and supportive personal relationships, a sense of belonging, and of contributing to and engaging in meaningful activities, have been found to be important [16].

Consequently, the main focus of personal recovery is not centered around a reduction of symptoms or medicinal intake but on the hope of finding ways to live well despite mental health challenges [17]. Or, as the director of the Boston Centre for Psychiatric Rehabilitation suggests, as a ‘deeply personal, unique process of changing one’s attitudes, values, feelings, goals, skills, and/or roles’ Recovery is not a scientific term, and therefore, there is not a specific theory behind recovery, but the term should rather be seen as a personal process one is going through. [18]. More recently, researchers have advocated a deeper social understanding of recovery, calling for an enhanced focus on the surroundings in which those with mental health issues find themselves, as well as an understanding of the everyday life of such patients in their particular social context [19]. The focus on the social dimension in recovery processes also calls for a broader understanding of the role of the outdoor environment in mental health services, especially in psychiatric services. We need to know, for example, whether the outdoor environment of a psychiatric facility can support and enable the formation of supportive relationships between patients with mental illnesses, or whether attractive outdoor surroundings can support links between patients’ treatment and their everyday life by providing facilities that encourage everyday outdoor activities, such as walk-and-talks, gardening, or exercise.

The aim of this scoping review is to provide an overview of the field of research dealing with the relationship between environment in which psychiatric wards are situated and the recovery of psychiatric patients. The scoping review will focus on the outdoor environment as a structural framework that can support the personal recovery of various patient groups, and where better recovery is the goal. The study aims to identify emergent themes that should be considered in the future design of new outdoor areas in psychiatric wards. The scoping review will focus on articles from 1990 and until the present day, since the current focus on personal recovery was first articulated 1990 [18].

## 2. Materials and Methods

The methodological procedures that were used to conduct this scoping review were based on Arksey and O’Malley’s five-stage framework for conducting a scoping review [20]. The five-stages are:Formulating the initial research questionIdentifying relevant studiesSelect relevant studiesCharting the dataCollecting, summarizing, and reporting the results.

The review is reported in accordance with the PRISMA guideline for scoping reviews [21]. The protocol was registered in the Open Science Framework in January 2021 and is available online through https://osf.io/5wy73/ (accessed on 31 December 2021).

### 2.1. Search Strategy

A pilot search in Prospero identified around 20 ongoing review articles dealing with the association between the outdoor settings and recovery. Only a limited number of these focus specifically on the importance of the environment for recovery among psychiatric patients. A few of them focused on children and adolescents [22] or on specific therapeutic programmes [23,24], which is not the main focus of this review. Therefore, the proposed review will differ from other attempted reviews by looking only at articles about the outdoor environment at psychiatry wards in relation to recovery among adults.

In the beginning of the process, search terms were identified in cooperation with a research librarian. The search terms were used in pilot studies before the final search to see if the terms were efficient. In January 2022, the final literature searches were carried out at University of Southern Denmark’s library in three online scientific databases: Embase, Medline and Psych-info (See Appendix A). In addition, two supplementary searches were conducted in Scopus and Google Scholar. In Google Scholar, the first 100 hits using the search terms were screened. To limit the search, we limited the search keywords shown in the Appendix B. In addition to these database searches, the first author screened the reference lists of the five studies included after the review process had been conducted.

We used the PEO (P = Population, E = Exposure and O = Outcome) question format for qualitative research questions [25]. The PEO model (See Table 1) is used for topics that deal with the relationship between exposure (exposure) and a health outcome (outcome). In order to broaden the search and investigate both population, exposure, and outcomes (PEO), the following research question was defined:

What is the connection between outdoor settings and recovery on psychiatric wards?

In the category population, we use the use the search words psychiatric* and psychiatry*. We are not interested in the diagnosis but in the fact that the patients are placed in a psychiatric ward.

In the category exposure, we used common words for outdoor areas, such as garden or more generally outdoor, but therapeutic and healing gardens have also been included since these are common words for outdoor areas focusing on patient well-being.

In the category outcomes, we used several synonyms for recovery. The effort of providing support for people undergoing a personal process of recovery is occasionally termed rehabilitation [26]. Therefore, rehabilitation is also used as a search term in this review. Other subheadings have also been added to get a deeper understanding of the concept.

We will include all studies regardless of methodology or design to provide a rich and descriptive overview of the current state of research, while acknowledging that as an emerging field it has yet no research cohesion.

### 2.2. Inclusion and Exclusion Criteria

We used several restrictions to narrow the search. The articles had to describe empirical studies, and the patients had to be above 18 years old. Otherwise, studies were excluded.

In addition, we used the following eligibility criteria: (a) the study only included research studies (in peer-reviewed journals) published between 1990 and the present day and (b) studies had to be conducted at mental wards and described in the English language.

### 2.3. Identifying Relevant Studies

Two reviewers, the first and second author, independently screened titles and abstracts from the databases for eligibility using the software Covidence. We also used Covidence to remove duplicates and manage full text versions of publications included. A total of 8138 references were identified through Medline, Embase, PsycINFO. After we had removed the duplicates, 5826 references remained. We excluded 5741 references as they did not meet the inclusion criteria and read the remaining 85 references in full text. After the full text screening, we excluded 80 references that had a study design, patient group or language that did not conform to our criteria. The final number of eligible references we included in this systematic scoping review was five.

The process of reference screening and study eligibility assessment is presented in the PRISMA flow diagram (see Figure 1).

### 2.4. Charting the Data

A data charting form was created based on the research question and the specific objectives guiding this review. Descriptive information of the studies, including the year and place of publication, population examined (Table 2) and diagnoses, recovery or rehabilitation, study design, and notion of mental health, was extracted and is summarized in Table 3.

### 2.5. Collecting, Summarizing, and Reporting the Results

Following the data extraction, the first author conducted a content analysis to identify which aspects the study should focus on, so we could identify emergent themes and collect and identify objectives and gaps in our understanding of the current state of research. A spreadsheet was developed to extract data from the five articles. Data extracted included year of publication, study location, study design and method, study population, aim, and variable categories, as well as main results and conclusions (Table 2 and Table 3). As data analysis progressed, the following topics were identified in each article: (a) study population characteristics, (b) recovery and well-being, (c) gardening and well-being, (d) contact with nature, and (e) the safe and supportive environment. The following discussion will be structured based on the themes that were identified.

## 3. Results

The review identified 5826 references after duplicate removal (see Figure 1), and a total of five articles were included in this review. Subsequently, we analyzed the articles to identify the overarching themes, i.e., a theme that clearly recurs in all five articles, and a number of sub-themes. The themes’ links to key elements of recovery are highlighted under each thematic section. This is to ensure a link between recovery and design.

### 3.1. Study Population Characteristics

All participants from the different studies had a mental disorder, such as mood disorders (major depression, bipolar disorder etc.), PTSD, schizophrenia, and personality disorders, although of varying severity. The studies including information about the age of the participants reported an age span of 20–72 years. In the studies that provided information on the gender of participants, a predominance of female participants was reported (27 out of 38 in total) [27,29,30,31].

### 3.2. Recovery and Well-Being

All studies examined different aspects of the relationship ranging from use of the environment to gardening activities as aspects in supporting the process of personal recovery. A common denominator in the studies is the way specific indicators are used to measure recovery. In general, self-reported ‘well-being’ serves as an overarching indicator when it comes to research on how environment and horticultural activities can affect mental health. In the following, similarities between how the different studies approach the concept of well-being will be outlined. Well-being was expressed in terms of mood, social relations, and autonomy. Social aspects as ‘a sense of belonging’ ‘togetherness’, ‘socialization and peer support’ were also found to be linked to increased well-being [29,30], and in the Swedish study an experience of meaningfulness was also identified [31]. Another aspect of well-being was the participants’ experience of their own autonomy, where independence and a sense of freedom were crucial in relation to their recovery process. In general, various aspects of autonomy presented a common dimension that some of the studies linked to well-being and mental health [28,29,30]. Three of the studies also associated reduced stress levels and the ability to relax with the experience of increased well-being gained from being in a natural environment [28,30,31].

### 3.3. Gardening and Well-Being

The examination of the participants’ mood in relation to being or working in garden environments was related to increased well-being in four of the five studies [28,29,30,31]. Only the Australian study did not talk about gardening. In both US studies using mixed methods, the value of gardening as a therapeutic tool in the recovery process was prominent. The studies were performed in two different psychiatric treatment facilities (in-patient and out-patient) and related to participants’ mood status. Both studies found that gardening provided feelings of being energized, of being less isolated, of happiness and of being able to enjoy. By supporting gardening, staff could make horticultural activity accessible to participants [27,29,30,31]. In addition to participants feeling less alone and isolated, a positive aspect of well-being found by four out of five studies that used gardening as a rehabilitation activity was that participants also built new social relationships [28,29,30,31].

### 3.4. Contact with Nature

Three studies also stressed that accompanying sensory experiences, such as smells, textures, temperatures, and colors, had a positive effect, allowing a distraction from inner thoughts, stress, and worries. Furthermore, the sensory experiences helped the participants to feel grounded and find focus [28,30,31]. The Italian study showed this both through qualitative interviews and through quantitative questionnaire. The Swedish study also pointed out how sensing nature and positive experiences with natural environments can be linked, resulting in an increased experience of relaxation when engaging with nature. Some of the other studies indirectly suggested that a varied physical setting is important, finding that increased well-being was supported by the opportunity for a variety of social and physical rehabilitation activities, for which the physical setting (e.g., a garden) is crucial.

### 3.5. The Safe and Supportive Environment

A consistent theme in many of the studies was the importance of a safe and supportive environment. Two of the studies found that such an environment required a clear architectural identity, a simple design with easily identifiable paths and entrances [28]. Another key finding relating to the creation of a safe and supportive environment was the importance of staff who could create a safe and encouraging atmosphere with positive group dynamics and increased trust. Furthermore, it was important that staff had the capacity to tailor and facilitate the various activities (gardening etc.) to the needs and prerequisites of the participants, so that they could make progress at their own pace. A common and pronounced theme in the Italian and Australian studies was how the physical environment played a role in patients’ recovery. Both studies found it important to offer patients a variety of physical settings with different features, so that patients could choose what kind of atmosphere/activity they wanted to be in (patient autonomy), e.g., if they wanted to be in a social setting or a more private setting. The Australian study (*n* = 20) also indicated that a lack of clear architectural identity can be perceived as a confusing space [27,28].

## 4. Discussion

The aim of this scoping review was to map the extent of existing empirical research to identify whether there is a connection between outdoor settings and personal recovery on psychiatric wards. At the same time, the scoping review also aimed to identify emergent themes that should be considered in the design of new environments on psychiatric wards.

Three common themes were identified during the review study: (1) gardening and well-being, (2) contact with nature, and (3) a safe and supportive environment. These themes address different aspects of design that can influence the recovery of psychiatric patients. The themes have strong connections to an already established theory, namely the stress reduction theory (SRT) proposed by Roger S. Ulrich [13]. Ulrich points at four factors that enhance stress restitution when incorporated into the design of therapy gardens and places with garden therapeutic qualities [32]. The factors are: (1) the feeling of control and access to protected or private environments, (2) social support, (3) physical movement and activity, and (4) natural distractions, i.e., positive distractions through contact with nature. Ulrich’s theory is grounded in general hospital care, but there is a strong likelihood that these factors could also apply to people admitted to psychiatry, since they are about people’s well-being in general. Therefore, research in this area can also be used in the design of new outdoor spaces around psychiatric centers. SRT is a well-established theory with clear guidelines. In contradiction, the concept of personal recovery is relatively unverified, and one could argue that the concept should be grounded in a more defined theory, before it can be taken into action. The anticipated outcome of recovery and SRT have common motives with a focus on well-being. While SRT focus on nature, recovery is centered around a personal process. A more thorough focus on nature could ease the process of personal recovery in the future. The reason why there are not more articles linking outdoor environments with recovery in psychiatric wards may be that the concept of personal recovery is relatively new in psychiatry, especially in a Danish context [33].

The articles used in the review were all published within the last seven years (2015–2020), even though we searched back to 1990. This indicates that the research area on recovery is growing, even though the number of publications is still small. The geographical spread across Europe, the United States, and Australia indicates that awareness of the issue is not geographically specific but extends throughout the Western world. Although the linkage between psychiatric wards, outdoor areas, and recovery remains fragile, we identified a large number of articles that described the positive effects of gardening [8] or time spent in nature [12], but these articles were all conducted outside psychiatric wards, even though they present relevant knowledge about the connection between well-being and outdoor settings.

### Strengths and Limitations

Some argue that scoping reviews can be criticized for a lack of thoroughness, as the approach is often mistakenly compared to systematic reviews [20]. In addition, the lack of quality assessment can be seen as a limitation of the scoping review. While this is generally not conducted in scoping reviews, it still adds potential bias to the review as studies with poor methodological quality are given the same weight as those whose methodological quality is good [34]. On the other hand, the greater flexibility of scoping reviews permits the inclusion of a broader field of methods and empirical design [34]. This broader approach can provide an overview of a larger field of research than is the case for systematic reviews with a narrower, in-depth aim. As the purpose of publication is to contribute an overview of existing knowledge, a scoping review was chosen as the method to obtain a knowledge base of the greatest breadth.

One strength of this scoping review was that the analysis permitted us to identify the connection between ‘well-being’ and outdoor activities. Furthermore, the literature search method was extensive and systematic, and it has provided a broad insight into the literature in this field. It was a strength that we searched in three scientific databases, which were all found to provide relevant articles relating to the purpose of the review. However, this review also has limitations. Most of the studies included in our scoping review used interviews with psychiatric patients, but the user groups were still rather diverse with different mental health conditions across each study. In addition, some findings were reported based on only few interviews, therefore the results should generally be interpreted with some caution. Finally, the results were based on studies from across the world with only two European studies. Therefore, the results may not be representative for a Danish context.

## 5. Conclusions

The purpose of the scoping review was to identify existing empirical research on the connection between outdoor settings and recovery processes at psychiatric wards. A total of only five articles was identified, indicating a lack of research within this specific topic. Nevertheless, the articles in general showed that outdoor settings can be related to the well-being of patients on psychiatric wards. The five articles suggest that in the field of health promotion, outdoor settings should be seen as a comprehensive resource for physical, mental, and social health and well-being, offering both contact with nature, varied spaces, and safe and supportive environments. However, further studies will have to be conducted to establish conclusively whether there is a connection between outdoor settings and recovery on psychiatric wards.

## Figures and Tables

**Figure 1 ijerph-20-02240-f001:**
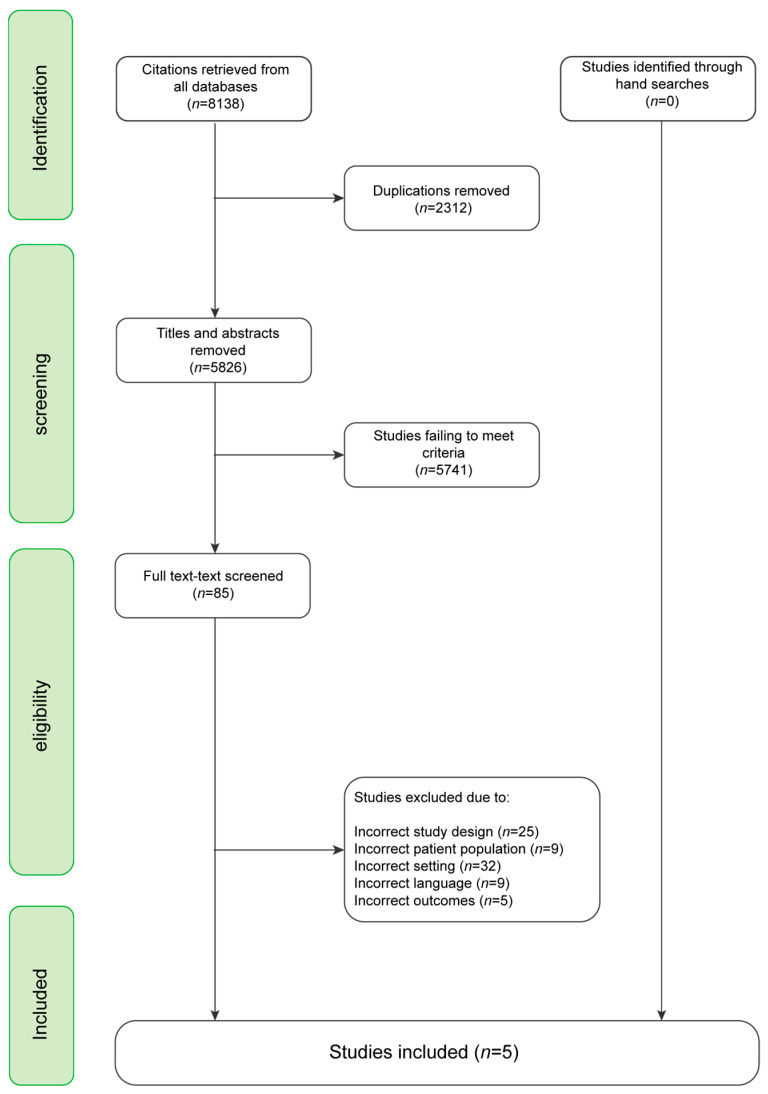
PRISMA flow diagram.

**Table 1 ijerph-20-02240-t001:** Search terms.

Population	Exposure	Outcomes
Psychiatr* department Psychiatr* ward Psychiatr* unit Psychiatr* hospital Psychiatr* diagnosis Mental disease Mental disorder Mental illness	Garden Land use Outdoor Green space Healing garden Horticultural therapy Nature Universal design Architecture Healing architecture Therapeutic landscapes Health care	Rehabilitated Recovered Recovery Recovering Rehabilitation Rehabilitate Rehabilitation process Rehabilitation program Rehabilitation treatment Resocialization therapy Resocialization

**Table 2 ijerph-20-02240-t002:** Overview of study characteristics.

- References - 1st Author, - Year of Publication—Country of Publication	Study Design Quantitative: Cross- Sectional/Longitudinal, Qualitative: Retrospective/ Prospective)	Study Population Characteristics: - N - Age - Sex Distribution
[27] **Donald et al., 2015, Australia**	Mixed methods Prospective: present study in existing environment. Semi-structured interview and focus group	Patients from a psychiatric department. N: 20 (9/11) Focus group: five women, four men. Semi-structured interview: six women, five men. Age: NA Sex: nine male, 11 female
[28] **Erbino et al., 2015,** **Italy**	Mixed methods Prospective: present study in existing environment. Case study: insights from patients, staff, and design professionals. Interviews and quantitative questionnaires with 100 participants (longitudinal	Patients with schizophrenia, personality disorders, severe depression N: 100 Age: 20 + Sex: NA
[29] **Smidl et al. 2017, USA**	Mixed methods Prospective: present study in existing environment. Semi-structured interviews and quantitative rating scales based on the Model of Human Occupation	Patients with mood disorders, schizophrenia, or schizoaffective disorder. Psychiatric diagnoses were concurrent with medical conditions. N: 20 Age: 26–72 years old. Sex: NA
[30] **Pieters et al., 2018, USA**	Qualitative Prospective study of the experiences of psychiatric inpatients. Semi-structured interview	Participants with major depressive disorder, anxiety disorder, PTSD N = 25 patients: Age: Average 28 years old. Sex: 14 women, 11 men.
[31] **Wästberg et al., 2020, Sweden**	Qualitative Prospective: participant perspectives and experiences during garden therapy. Narrative individual interviews	Participants: Able to participate in garden therapy. Sick due to common mental disorders. Two different social groups. N: 8 Age: 32–61 years old. Sex: one male, 7 female

**Table 3 ijerph-20-02240-t003:** Overview of the results.

References	Aim/Approach	Treatment, Setting and/or Activity	Indicators Related to Mental Health	Results/ Conclusions
[27]	To investigate the role of places in mediating recovery by addressing the environmental impact of hospital-based settings	A Low Dependency Unit and secure High Dependency Unit. A mixture of recreational and therapeutic activities	Well-being	The importance of staff; lack of clear architectural identity resulting in confused or confusing space; and limited amenity due to poor architectural design
[28]	To create a master plan for healing gardens, applied to a case study. How contact between human and nature/cultivating a healing garden can improve mental and physical well-being	Natural elements, a variety of spaces, lawns, flowerbeds, tree-lined avenues, shrubbery. Garden activity, rehabilitation program	Mental and spiritual well-being Stress reduction for patients, staff, and relatives Increased autonomy (patients)Reduction in care cost Improved mood Quality of life	Designing of green healing areas requires consideration of key issues such as: contact with nature, patient autonomy and ease of orientation, patient safety and comfort, possibility of choosing between places and functions
[29]	To explore the value of a therapeutic gardening project as part of the psychosocial recovery curriculum provided at a community mental health centre.	A community mental health centre. Activities were organized in four phases: ‘planning, construction of raised-bed gardens, maintenance of the gardens, and harvest.	Pride and self-worth Happiness Connecting past and present Socialization, peer support Decreased isolation Personal and social responsibility Need of less structure, encouragement	The Recovery Garden project provided an opportunity for clients with severe mental illness to demonstrate choice, initiative, self-direction, and collaboration within a new healthy occupation
[30]	To explore and describe the experiences of gardening in an acute psychiatric inpatient setting in the participants’ own words. A way to implement the Recovery Model to identify with and build skills to work towards finding new meaning and occupations	Weekly garden group, initiated by occupational therapists and implemented as an intervention by the unit’s multidisciplinary team	Ability to concentrate and relax Feeling calm Cognition Improved mood Distractions Productivity Decreased feeling of being isolated Ability to take on new information Sense of freedom Feeling energized Sense of belonging	Key elements to conducting a gardening group to ensure optimal results: format and structure of the group, activities such as ‘turning the soil,’ ‘watering…’, staff involvement, social interactions, experiences with nature, sensory perceptions, experiences of being, being grounded, finding focus
[31]	To serve as a complement to regular out-patient psychiatric care for clients with CMD to enhance their well-being. The study investigates whether and why participants perceive garden therapy as meaningful	An out-patient psychiatric clinic in Sweden Various garden activities	Senses of restoration Being in the moment Senses of belonging Feeling supported Togetherness Stress relief Feeling safe and accepted Quality of life	Garden therapy perceived as being meaningful, but different needs and prerequisites influenced what specific components in the garden therapy they perceived as meaningful

## Data Availability

Not applicable.

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
