# Peer review of "The Importance of the Outdoor Environment for the Recovery of Psychiatric Patients: A Scoping Review"

_ijerph, 2023, doi:10.3390/ijerph20032240_

Round 1

Reviewer 1 Report

More strength and limitations is needed to recognise the importance and the gap of the research.

1. You had mentioned that the study was to focus on the Scandinavian countries, but your data was collected outside of those countries anyway. What is relevant about that?
2. If you decided to do a review paper, you should clearly state the theory behind your topic.
3. An understandable figure or table is needed to illustrate the results.

Author Response

Dear reviewer

Thank you very much for your thorough comments on the article. I have tried to make revisions to accommodate your comments and I hope you find the improvements fulfilling. I look forward to your reply

Please see the attachment and text below.

  1. More strength and limitations is needed to recognise the importance and the gap of the research. More information about strength and limitations has been added.
  2. You had mentioned that the study was to focus on the Scandinavian countries, but your data was collected outside of those countries anyway. What is relevant about that? I cannot find the place where I mention the study should focus on the Scandinavian countries, but I have mentioned the point as a limitation.
  3. If you decided to do a review paper, you should clearly state the theory behind your topic. The overarching topic of the article is recovery. Recovery is not a scientific term, and therefore, there is not a specific theory behind recovery, but the term should rather be seen as a personal process one is going through. Ulrich has made the theory of SRT, which are also guiding design today. The theory is described in the introduction and used again in the discussion.
  4. An understandable figure or table is needed to illustrate the results. Appendix 3. is made simpler and more vigorous explaining the basic information about the used methods and participants. Appendix 4. gives and overview of the results.

Reviewer 2 Report

Dear Editor and Authors,

Thank you for giving me the opportunity to review this manuscript. The authors have performed a scooping review about the importance of the out-door environment for the recovery of the psychiatric patients. This area is highly interesting but challenging area to research. Therefore, I was not surprised that so many papers were exclude and did not meet the inclusion criteria. The scoping review is well performed and follows the procedure of systematic reviews described in the literature and as I understand thew gray literature was not included. The references are relevant and the findings are well structured.

There are only a few methodological aspect that I would like to raise before I can recommend the paper for publication.

The table over included papers needs revision, it’s very hard for the readers to grasp the content, please revise and clarify. It would be helpful to know the methods used in the included articles in order to understand the findings better, what kind of qualitative method? And mixed methods should be clarified questionnaires and interviews or??

Where there any quantified measures used in the included studies?  In that case it would be helpful to have them presented. The process of analysis should be explained in more carefully, how did you analyzed the data?

The findings are interesting, however, it would valuable if the authors provided more information related to each study where the findings are synthesized/summarized included.

For example “ the Australian study with qualitative design (n=7) stressed out-door environment strengthened ….”

The discussion section is small and could preferably be extended. Please also insert / discuss the findings from a theoretical perspective.

Author Response

Dear reviewer

Thank you very much for your thorough comments on the article. I have tried to make revisions to accommodate your comments and I hope you find the improvements fulfilling. I look forward to your reply.

See attachment and text below.

  1. The table over included papers needs revision, it’s very hard for the readers to grasp the content, please revise and clarify. I agree on your comment, so now appendix 3. is made simpler and more vigorous explaining the basic information about the used methods and participants. Appendix 4. gives and overview of the results.

  1. It would be helpful to know the methods used in the included articles in order to understand the findings better, what kind of qualitative method? And mixed methods should be clarified questionnaires and interviews or? Where there any quantified measures used in the included studies?  In that case it would be helpful to have them presented. It is mentioned in appendix 3. now.

  1. The process of analysis should be explained in more carefully, how did you analyzed the data? I have tried to elaborate more over the subject in section 2.5. I decided not to use nvivo, since we only had five articles, so I just used a spreadsheet to organize the themes.

  1. The findings are interesting, however, it would valuable if the authors provided more information related to each study where the findings are synthesized/summarized included. For example “ the Australian study with qualitative design (n=7) stressed out-door environment strengthened ….” I've added more info in a few places, but I think it's too much if you have to do it everywhere. Then the text will not be pleasant to read.

  1. The discussion section is small and could preferably be extended. Please also insert / discuss the findings from a theoretical perspective. More information about strength and limitations has also been added about strength and limitations in section 4.1. Ulrich has made the theory of SRT, which are also relevant today. The theories are described in the introduction and used again in the discussion. There is added more text about the link between recovery and SRT. It can be difficult to connect a well-established theory with a relatively new process.
